# Antioxidants in Traditional Mexican Medicine and Their Applications as Antitumor Treatments

**DOI:** 10.3390/ph16040482

**Published:** 2023-03-23

**Authors:** Karen M. Soto, José de Jesús Pérez Bueno, Maria Luisa Mendoza López, Miguel Apátiga-Castro, José M. López-Romero, Sandra Mendoza, Alejandro Manzano-Ramírez

**Affiliations:** 1Centro de Investigaciones y de Estudios Avanzados del I.P.N., Unidad Querétaro, Querétaro 76230, Mexico; 2Centro de Investigación y Desarrollo Tecnológico en Electroquímica, S.C., Parque Tecnológico, Querétaro-Sanfandila, Pedro Escobedo, Santiago de Querétaro 76703, Mexico; 3Tecnológico Nacional de México, Instituto Tecnológico de Querétaro, Av. Tecnológico s/n, Esq. Mariano, Escobedo Colonia Centro, Santiago de Querétaro 76000, Mexico; 4Centro de Física Aplicada y Tecnología Avanzada, Universidad Nacional Autónoma de México, A.P. 1-1010, Querétaro 76230, Mexico; 5Research and Graduate Program in Food Science, Universidad Autónoma de Querétaro, Querétaro 76010, Mexico

**Keywords:** traditional medicine, Mexican plants, antioxidant, antitumor

## Abstract

Traditional medicine in Latin America and mainly in Mexico represents an essential alternative for treating different diseases. The use of plants as medicine is the product of a rich cultural tradition of the indigenous peoples, in which a great variety of species are used for the treatment of gastrointestinal, respiratory, and mental diseases and some other sicknesses; the therapeutic efficacy that they possess is due to the properties that derive from the active ingredients of plants principally antioxidants, such as phenolic compounds, flavonoids, terpenes, and tannins. An antioxidant is a substance that, at low concentrations, delays or prevents substrate oxidation through the exchange of electrons. Different methods are used to determine the antioxidant activity and the most commonly used are described in the review. Cancer is a disease in which some cells multiply uncontrollably and spread to other parts of the body, a process known as metastasis. These cells can lead to the formation of tumors, which are lumps of tissue that can be cancerous (malignant) or noncancerous (benign). Generally, the treatment of this disease consists of surgery, radiotherapy, or chemotherapy, which have side effects that decrease the quality of life of patients, so new treatments, focusing on natural resources such as plants, can be developed. This review aims to gather scientific evidence on the antioxidant compounds present in plants used in traditional Mexican medicine, specifically as antitumor treatment in the most common cancer types worldwide (e.g., breast, liver, and colorectal cancer).

## 1. Introduction

In recent years, cancer has become one of the most critical diseases in the world due to its high incidence and mortality rate. Based on the reported data, in 2020 more than 19.3 million new cancer cases were diagnosed and reported, leading to approximately 10 million deaths [1]. Cancer is characterized by an uncontrolled proliferation of abnormal cells and aberrant recognition of the immune system; it can occur in different body organs, such as the breast, prostate, colon, lung, lymph, blood, brain, and kidney. Cancer has various causes, but the most common are mutations or alterations in the expression patterns of proto-oncogenes, tumor suppressor genes, and genes involved in DNA repair, caused by environmental factors, such as exposure to radiation and pollutants and unhealthy lifestyles, including lack of physical activity, unbalanced diet, smoking [2,3,4]. The treatment depends on the type of cancer, the stage of the disease, and the patient’s ability to withstand the given therapy. However, the treatment is generally based on surgery and chemo- or radiation therapy to kill the cancer cells. Surgery causes functional deficiencies or esthetic discomfort; on the other hand, chemo- or radiotherapy is expensive and causes many side effects, including hematological toxicity, vomiting, appetite loss, delirium, diarrhea, fatigue, hair loss, pain, nausea, myelosuppression, and neurological, cardiac, pulmonary, and renal toxicity. These side effects reduce the quality life of the patients, and they have limited anti-cancer activity. Therefore, new treatment alternatives are sought, which minimize side effects and present excellent teratogenic effects against cancer cells [5].

Approximately 60% of principal drugs used in cancer treatments are produced with natural products, principally extracted from plants; medicinal plants have been used for many years in developing countries as the primary source of medical treatment, and developing nations are utilizing the benefits of naturally sourced compounds for therapeutic purposes [6,7]. In Mexico, ancient civilizations such as the Maya and the Aztecs developed some uses for medicinal plants, which were enriched by the conquering cultures. Currently, Mexico is one of the countries with the most significant number of plant species (31,000), and plants are also used as condiments and for ornamental, medicinal, and aromatic purposes. More than 3350 plants are part of the medicinal flora and play an essential role in public health in local communities [8]. Plants’ effects on different diseases are due to the compounds that they contain, mainly antioxidants such as phenolic acids, phenols, tannins, and anthocyanins [9].

In the present review, we present a summary of the literature that shows results from studies of the antitumor effects of antioxidants in Mexican plants used in traditional medicine.

## 2. Antioxidants

Antioxidants are natural or manufactured compounds that interact with free radicals and can neutralize them by donating electrons; they have been widely used as food additives to prevent lipid peroxidation and as adjuvants in reducing the risk of some diseases such as cancer and many chronic degenerative diseases such as coronary heart diseases, cardiovascular diseases, and aging [10]. Antioxidants can be classified as endogenous, exogenous, or synthetic; the first is produced naturally by the human body and consists of glutathione peroxidase, superoxide dismutase, and catalase, while non-enzymatic antioxidants are uric acid, lipoic acid, bilirubin, glutathione, and melatonin. Exogenous antioxidants are found in foods such as vegetables, fruits, and plants and consist of carotenoids, phenolic compounds, vitamins E, A, and C, and natural flavonoids or other compounds. Synthetic antioxidants correspond to petroleum-based compounds such as butylated hydroxytoluene (BHT), butylated hydroxyanisole (BHA), propyl gallate (PG), and tert-butylhydroquinone (TBHQ) [11,12].

### 2.1. Antioxidant Activity and Capacity

Antioxidant activity refers to the constant reaction rate between a specific antioxidant and an oxidant. At the same time, capacity measures the amount of a given free radical scavenged by a sample. Many methods exist with highly sensitive and automated antioxidant capacity quantification technologies; some quantify the scavenging activity against certain types of free radicals or reactive oxygen species, reducing powder, or metal chelation [13]. Some of the most used are described below.

#### 2.1.1. 2,2-Diphenyl-1-Picrylhydrazyl (DPPH) Radical Scavenging Assay

The DPPH technique is based on the inactivation of the DPPH radical through the donation of electrons by antioxidants; the DPPH radical is a violet chromophore that changes to a yellow color when reduced; this change is measured at an absorbance of 527 nm. It is a simple technique that only requires the purchase of the radical and a spectrophotometer. Results are generally reported as mg equivalents of Trolox, the compound used as standard [13].

#### 2.1.2. 2,2′-Azinobis (3-Ethylbenzothiazoline-6-Sulphonic Acid (ABTS^•+^)

This method quantifies the discoloration of the ABTS^•+^ radical due to its reduction to ABTS^•+^ by the action of antioxidants. The cationic radical ABTS^+^ is a bluish-green chromophore that absorbs at a wavelength of 734 nm and is generated by an oxidation reaction of ABTS^•+^ (ammonium 2,2′-azino-bis-(3-ethylbenzothiazoline-6-sulfonate) with potassium persulfate Thus, the degree of discoloration as a percentage of inhibition of the ABTS^•+^ radical is determined as a function of the concentration. Antioxidants can neutralize the radical cation ABTS^•+^ by either direct reduction via electron donation or by radical quenching via hydrogen atom donation. The antioxidant structure and pH of the medium generally determine the balance of these two mechanisms. The results are generally reported as TEAC (Trolox equivalent antioxidant capacity) [14].

#### 2.1.3. Ferric Reducing Antioxidant Power Assay (FRAP)

FRAP is a colorimetric method that evaluates the ability of antioxidant compounds to reduce ferric iron (Fe^3+^) present in a complex with 2,4,6-tri(2-pyridyl)-s-triazine (TPTZ) up to the ferrous form (Fe^2+^); this method is carried out under acidic conditions (pH 3.6). In the presence of antioxidants, the ferric form of the compound iron-tripyridyl-triazine (Fe^3+^-TPTZ; yellow color) is reduced to the ferrous state (Fe^2+^-TPTZ; blue color). The Fe^2+^-TPTZ compound produces an intense blue coloration with a maximum absorption of 593 nm. The FRAP method is, therefore, a method that does not evaluate the neutralizing capacity of free radicals of the sample studied but rather its reducing capacity by electron transfer, contrary to the ABTS and DPPH methods [15].

#### 2.1.4. Oxygen Radical Absorbance Capacity ORAC Assay

The assay measures the oxidative degradation of a fluorescent molecule (fluorescein) after it has been mixed with free radical generators such as azo compounds (2,20-azobis(2-amidinopropan) dihydrochloride). The azo derivatives are considered to produce peroxyl radicals by heating, which damages the fluorescent molecule, resulting in the loss of its fluorescence. Antioxidants protect the fluorescent molecule from oxidative degeneration. The degree of protection is quantified using a fluorometer. If an antioxidant is added to the test, the antioxidant reacts with the ROS, which delays the oxidation of fluorescein. Trolox and Vitamin E are used as standard [16].

#### 2.1.5. Electrochemical Methods

Conventional methods such as DPPH, ABTS, and FRAP present different disadvantages because when they are monitored by UV spectroscopy, there may be factors that give an erroneous reading, such as interferences with chromophoric compounds present. Electrochemical methods based on the cyclic, differential pulse, square wave voltammetry, and coulometry represent an essential alternative; these techniques are sensitive, rapid, and simple, and they can directly measure the number of electrons transferred by an antioxidant. An electrochemist station and the corresponding electrodes are necessary to carry out these techniques [17,18].

### 2.2. Mexican Plants with Antioxidant Activity

Mexican plants represent a vast reservoir of compounds with excellent properties, capable of helping cure different diseases; for this reason, they have been used in traditional medicine since ancient times because they are cheap, widely available, and considered non-toxic since they are natural (it is important to emphasize that some plants can be toxic, depending on the dose consumed). The National Commission for the Knowledge and Use of Biodiversity (Conabio) [19] mentions the registry in the Mexican Institute of Social Security of 3000 species (of the 4000 that are estimated to exist in Mexico) of plants with medicinal attributes. These 4000 species represent 15% of the total Mexican flora (approximately 50,000 species). It specifies that only 5% of the total number of plants with medicinal attributes have been pharmacologically analyzed, and only 250 species are commercialized. Of these, 85% are extracted from the wild without sustainable management plans, and 80% of the Mexican population has used them [6,9,20].

Table 1 shows the values of antioxidant activity of some of the most used medicinal plants in Mexico for different sicknesses. The effect of the plants is related to a mixture of various compounds such as alcohols, esters, aldehydes, ketones, carbohydrates, terpenes, polyphenols, anthocyanins, tannins, and phenolic acids [21].

The antioxidant activity of plants is related to the different compounds found in all their parts, from roots, stems, leaves, and flowers (Figure 1). Generally, for measurement, extracts are made with different solvents, mainly ethanol and methanol. Some examples of this are the following: *Annona muricata* Linn., commonly known as soursop, is a plant used in traditional Mexican medicine to treat hypertension, diabetes, stomach pain, fever, parasitic infections, vomiting, cancer, and others conditions. Ethyl leaf fractions showed an excellent antimicrobial activity of 3964 ± 53 μmol Trolox eq g^−1^ in the ORAC test, and the principal components identified were chlorogenic and caffeic acids, procyanidins B2 and C1, (epi)catechin, quercetin, quercetin-hexosides, and kaempferol, a compound known as a potential antioxidant [22]. *Castilleja tenuiflora* Benth. (Orobanchaceae), commonly known as “Indian paintbrush,” is distributed in mountainous areas of the southern USA and Mexico; this plant is frequently used in the treatment of symptoms of various cancers, coughs, inflammation, and gastrointestinal disorders; its activity may be related to the biological activities of its secondary products, including iridoid glycosides, phenylethanoid glycosides), and flavonoids, compounds that allow it to possess an antioxidant activity of 161.74  ±  10.06 μmol Trolox/g dry weight in ABTS free-radical scavenging [23]. *L. graveolens*, Mexican oregano, has been commonly used in traditional medicine as a solid auxiliary against specific ailments linked to microorganism infections or inflammatory processes. The main components of oregano extracts have been associated with the capability of bacterial control, including against antibiotic-resistant strains; methanolic extracts of oregano showed an IC_50_ (inhibitory concentration 50) of 207.96 ± 1.43 in the DPPH assay, and the principal compounds are thymol, carvacrol, rosmarinic acid, naringenin, and some other phenolic compounds [24,25]. *Tithonia diversifolia* (Hemsl.) A. Gray (Asteraceae) is a plant native to Mexico and grows in parts of Africa, Australia, Asia, and America. Its flower extracts are traditionally used in treating diabetes, diarrhea, menstrual cramps, malaria, hematomas, hepatitis, hepatomas, and wounds. Flower extracts had an IC_50_ value of 265 μg/mL in the DPPH assay; different compounds were identified in these extracts, such as flavonoids, alkaloids, coumarins, tannins, and saponins [26,27]. *Tagetes lucida*, commonly named pericon, Mexican tarragon, and Mexican mint marigold, is an herbaceous, perennial, and endemic plant in Mexico and Guatemala. The infusion of this plant is widely used in traditional medicine to cure gastrointestinal disorders. It is reported to possess bactericidal and platelet antiaggregant activity and inhibitory effects on smooth-muscle contraction. The IC_50_ of the aqueous extract was 6.4 μg, and compounds such as gallic acid, quercetin, kaempferol, and rutin were present [28,29]. *Tagetes erecta* L. is widely commercialized as an ornamental plant for Dead Day in Mexico due to its showy orange and yellow flowers. In Mexico, it is known as the cempasuchil flower. The infusion of its flowers is used in traditional medicine to treat gastrointestinal diseases such as dyspepsia and diarrhea; it presents antimicrobial, antifungal, antioxidant, anti-inflammatory, and analgesic effects related to the presence of flavonoid compounds. In the ABTS assay, the antioxidant activity was 401.47 ± 3.35 mg TE/g of dry weight, and the principal components were phenolic acids such as caffeic acid, ellagic acid, coumaric acid, chlorogenic acid, protocatechuic acid, and rosmarinic acid, and flavonols such as rutin, apigenin, and kaempferol [8].

**Table 1 pharmaceuticals-16-00482-t001:** Antioxidant activity of different Mexican plants used in traditional medicine.

Plant	Common Name	Part	DPPH (mg TE/g)	ABTS (mg TE/g)	TPC (mg GAE/g)	Reference
*A. muricata*	Guanabana	Leaves	28.1 ± 4.4	--	79.4 ± 6.4	[22]
*Castilleja tenuiflora*	Cola de borrego	Aerial parts	49.91 ± 1.80	112.85 ± 7.03	30.58 ± 2.39	[23]
*L. graveolens*	Oregano	Leaves	81.00 ± 5.0	--	270.25 ± 4.1	[25]
*Tithonia diversifolia*	Sunflower	Leaves	217.2 ± 8.70	--	14.6 ± 0.64	[26]
*Bougainvillea buttiana* (Var. Orange and Rose)	Bugambilia	Flowers	1683.6 ± 143.3	--	29.5 ± 0.05	[30]
*Piper auritum*	Yerba santa	Leaves	--	14.82 ± 2.88	6.79	[31]
*Justicia spicigera*	Muicle	Leaves	880 ± 94	8480 ± 378	8520 ± 497	[32]
*Tribulus terrestris*	Abrojo	Aerial	~35	--	250 ± 1.17	[33]
*Parthenium argentatum* A. Gray	Guayule	Leaves	~21.3–27.4	--	~16–27	[34]
*Parmentiera aculeata* Kunth	Cuajilote	Fruit	~160	--	~1980	[35]
*Arctostaphylos pungens*	Pingüica	Fruit	6214 ± 132	8465 ± 124	323.4 ±5.6	[36]
*Thymus vulgaris*	Tomillo	Leaves	IC_50_ 13.4 μg/mL	IC_50_ 40.03 μg/mL	~256	[37]
*Eryngium carlinae*	Hierba del sapo	Inflorescence	45.02 ± 0.31	197.2 ± 75	4.32 ± 0.02	[38,39]
*Bixa orellana* L.	Achiote	Seed	17.42 ± 0.45	--	62.08 ± 2.21	[40]
*Acacia farnesiana*	Huizache	Aerial	89 μmol TE/g ORAC	3.4 g TE/g FRAP	565	[41]
*Taraxacum officinale*	Diente de Leon	Leaves and flowers	0.950 ± 0.002	1.132 ± 0.012	0.535 ± 0.033	[42]
*Tagetes erecta* L.	Cempasuchil	Flowers	401.47 ± 3.35	843.92 ± 4.44	108.71 ± 1.13	[8,43]
*Arnica montana*	Arnica	Roots	--	--	116.9 ± 1.0	[44]
*Ruta graveolens* L.	Ruda	Leaves	67.59 ± 0.98	--	30.19 ± 0.16	[45]
*Tagetes lucida* Cav	Pericon	Flower	---	--	12.7 ± 0.1	[46]
*Anchusa officinalis* L.	Lengua de Buey	Flower	57.04 ± 1.08	--	104.03 ± 0.63	[47]
*Passiflora incarnata*	Pasiflora	Flower	IC_50_ 31.92 μg/mL	--	--	[48]
*Acalypha wilkesiana*	Chirrite	Leaves	IC_50_ 53.49 μg/mL	--	~50	[49]

TPC, total phenolic compounds; GAE, gallic acid equivalents; IC_50_, concentration that reduces the activity by 50%; TE, Trolox equivalent; (--) not shown.

## 3. Plants in the Treatment of Breast Cancer

Breast cancer (BC) is one of the most common cancers and is considered one of the leading causes of death in women. In 2020 it was estimated that 2,262,419 new cases were diagnosed, and 684,996 were reported. BC does not only occur in women; men also suffer from this type of cancer; in 2018, men’s mortality rates increased dramatically. Generally, BC is diagnosed early when the tumor can be extracted; however, 20–30% of patients suffer metastasis in different body parts. Different factors increase cancer risk, including genetic, reproductive, and lifestyle factors, hormonal imbalances, and other factors such as breast density, age, and change in circadian rhythm. The most common treatments for BC are chemotherapy, radiation, and principally surgery, where the breast is extracted entirely [50,51]. Figure 2 shows some plants used for breast cancer treatment.

*Hypericum perforatum*, known as San Juan in Mexico, is the most studied Hypericum species. It is known for its pharmacological antidepressant activities and antiviral and antibacterial properties. The aerial extracts’ principal component contains antioxidants such as quercetin, kaempferol, hyperiside, hypericin, catechins, apigenin, luteolin, and chlorogenic, caffeic, vanillic, p-hydroxybenzoic, and ferulic acids, and some condensed tannins. Abbas et al., 2016, studied the cytotoxic effect of hypericin, the principal component of H. perforatum, in the MCF-7 cell line derived from a patient with metastatic breast cancer. LD_50_ (lethal dose 50) of the hypericin in MTT assay was 5 μg/mL; the principal mechanism is the increment in the p53 expression and a decrease in bcl2, guardian cells in front of tumor formation, and anti-apoptotic genes [52,53].

*Cestrum nocturnum* L., lady of the night, or night jessamine, is an evergreen shrub from the family Solanaceae that grows in tropical and sub-tropical regions worldwide. In Mexico, it grows in the center and the Yucatan peninsula, where it is used in traditional medicine for its cytotoxic, hepatoprotective, and antitumor effects. Kumar et al., 2022, studied the compounds related to the cytotoxic-effect extracts. The results revealed that MFLCN (methanolic fraction obtained from ethyl acetate extract of leaf of Cestrum nocturnum) contained 33 flavonoids (4 in positive and 29 in negative ion modes) of different classes; some examples were kaempferol, apigenin, baicalin, quercetin, hyperoside, and vitexin. Furthermore, the cytotoxicity effect of the aqueous extract in breast carcinoma cell MCF-7 was investigated by Rashed et al. in 2018, getting a value of LD_50_ of 55.28 ± 4.87 μg/mL; the authors attributed the effect to the presence of large amounts of kaempferol, an antioxidant compound that has been highly related to cell death in different cancer cell lines [52,54].

*Lophophora williamsii* (LW), also known as peyote, devil’s root, dumpling cactus, or sacred mushroom, is a spineless, tufted, blue-green, button-like cactus. It grows wild in the center and north of Mexico, Nayarit, Querétaro, San Luis Potosí, Durango, Zacatecas, Nuevo León, and Baja California Norte, among other areas. It is used as an analgesic, stimulant, and antibiotic for its hallucinogenic properties. Franco et al., in 2017, studied the in vitro tumor cell toxicity in the MCF7 breast cancer cell line and another type of cancer (lymphoma, fibroblastoma). MCF7 was found to be the most sensitive cell line with a dose-response effect, which at 18 μg/mL, reduced cell viability by up to 1.3%; the study demonstrated that peyote extract was capable of stimulating lymphocyte proliferation and killing tumor cells [55,56].

*Cordia boissieri*, commonly known as Mexican anacahuita, is an ornamental shrub or tree up to 30 ft, with large, soft, dark leaves and large, showy, trumpet-shaped white flowers with yellow throats that are sometimes described as looking like crepe paper or chiffon. Different parts of this plant are used in traditional medicine; for example, the roots are used as emollients, and the flowers are used to treat coughs and colds, while the fruits and leaves are used to alleviate rheumatism and pulmonary illness. The components related to these effects depend on the area from which the extracts are obtained; the fruits are generally rich in kaempferol and hydroxy aldehyde, and in the flower extract, mainly (−)-spathulenol (19.1%) and (E)-caryophyllene (16.2%) have been identified [57]. Viveros-Valdez et al., in 2016, studied the cytotoxic effect of fruit extracted of *C. boissieri* against the MCF7 cancer breast cell line and obtained an IC_50_ of 310 ± 42 μg/mL; the cytotoxic effect is moderate compared to that of other fruit extracts, but it could be a good source of antioxidant compounds with an alpha-glucosidase inhibitory effect and antitumor effect in specific cancer types [58].

*Bursera fagaroides* var. fagaroides are a wild tree endemic to Mexico, known as aceitillo, copal, or cuajiote amarillo; its stem bark and exudates are both used in folk medicine to treat cuts and tumors; various studies have demonstrated its amoebicidal, immunomodulatory, and antitumoral activities. Peña-Morán et al., in 2016, studied the cytotoxicity effect of four compounds isolated from *B. fagaroides*, namely 51-demethoxy-peltatin-A-methylether (1), acetylpodophyllotoxin (2), 51-demethoxydeoxypodophyllotoxin (3), and 81-dehydroacetylpodophyllotoxin (4), compound 3 showed the lowest IC_50_ against MCF-7= 0.04 ± 0.01 μM; MDA-MB-23 = 0.145 ± 0.04 μM; and MCF-10A = 0.09 ± 0.009 μM [59,60].

## 4. Plants in the Treatment of Liver Cancer

Liver cancer is the fourth leading cause of cancer-related mortality worldwide, despite the liver being the sixth most common site of primary cancer. Hepatocellular carcinoma (HCC) accounts for 80–90% of primary liver cancers, and cholangiocarcinoma (CCA) accounts for 10–15%. Angiosarcoma and pediatric hepatoblastoma account for a relatively small proportion. Risk factors include hepatitis B virus, hepatitis C virus, fatty liver disease, alcohol-related cirrhosis, smoking, obesity, diabetes, iron overload, and various dietary factors because the liver plays a key role not only in the metabolism of macronutrients but also in detoxification and hormone production. Thus, the diet has measurable biological impacts on key pathways hypothesized to be involved in liver cancer risk [61,62,63]. One of the most important treatments for liver cancer is surgical resection or liver transplantation, depending on whether the patient is a suitable transplant candidate. However, most patients with liver cancer are diagnosed late, thereby excluding the patients from that treatment. In addition, chemotherapy and radiotherapy are generally ineffective for this type of cancer [64]. Table 2 summarize some of the plants used for the treatment of liver cancer.

*Brickellia cavanillesii* is a perennial herb endemic to Mexico and known commonly as “prodigiosa”, “atanasia amarga,” or “hamula”, among other names. It is generally commercially found in herbal stores and is used as a treatment for ulcers, dyspepsia, and diabetes (as a cheaper alternative to insulin). The chemical composition of *B. cavanillesii* plants consists of a glycoside named Brickellin, resin, essential oil, fat, tannin, coloring material, gum, starch, chlorophyll, and mineral salts. Eshiet et al., 2014, reported the presence of different antioxidant compounds in the ethanolic extract of *B. cavanillesii;* some of these were phenol, 2-methoxy-4-(1-propenyl); benzene, 1-(1, 5-dimethyl-4-hexenyl)-4-methyl-; phenol, 2-methoxy-; benzaldehyde, 3-hydroxy-4-methoxy-; 11, 13-eicosadienoic acid, methyl ester; maltol; phenol; and hydroquinone [65,66]. Viñas and Smith, in 2014, reported the potential cytotoxicity of Prodigiosa herbal tea on HepG2 cells (hepatocellular carcinoma from the liver tissue of a 15-year-old woman); they observed a dose- and time-dependent effect at a concentration of 200 mg/L, exhibiting excellent toxicity after 48 h that was associated with the regulatory mechanisms on GLUT2 (glucose transporter) expression and cDNA levels in HepG2 cells [67].

Maize (*Zea mays* L.) is one of Mexico’s primary food sources and is used to manufacture dry masa flours, chips, and tortillas. Different species of maize exist in Mexico, but native blue-pigmented maize has attracted extensive attention due to its nutraceutical perspective; the response of the blue color is the anthocyanins and other phenolic compounds mainly associated with the pericarp and the monolayered aleurone. Anthocyanins are essential nutraceutical compounds with high antioxidant capacity and are used to prevent or treat chronic degenerative diseases such as atherosclerosis, aging, diabetes, hypertension, inflammation, and cancer [68,69]. Urias-Lugo et al., in 2015, evaluated the cytotoxic effect of acidified and non-acidified extracts of blue maize on Hep2 cells (liver cancer); they were able to observe that at a concentration of 5 mg/mL, the acidified extract was capable of reducing cell viability by up to 70%. Furthermore, they observed a strong correlation between Cy3-Glu (cyanidin-3-O-glucoside) concentration with the cytotoxic effect [69].


**Table 2 pharmaceuticals-16-00482-t002:** Mexican medicinal plants with cytotoxic activity against different cell lines.

Plant	Part of the Plant	Solvent	Cellular Line	IC_50_ μg/mL	Reference
*Rhoeo discolor*	Leaves	Methanol	HeLa	70 ± 3.2	[70]
	Roots			67 ± 1.4	
*Lophocereus schottii*	Stem	Ethanol	L5178Y	7.8	[71]
*Annona muricata* Linn	Leaves	Ethanol	4T1	79.2 ± 0.2	[72]
*Annona squoamosa*	Seed	Ethanol	PC-3	13.08	[73]
			SiHa	16	
*Barringtonia racemosa*	Fruit	Methanol	MCF-7	57.61 ± 2.24	[74]
*Hibiscus sabdariffa*	Flower	Methanol	MCF-7	112.10 ± 3.97	[74,75]
*Justicia spicigera* Schltdl	Leaves	Ethanol	HeLa	17	[76]
*Tagetes lucida* Cav.	Flowers	Water	Calu-1	100	[46]
			HepG2	270	
*Dioon spinulosum*	Leaves	Ethanol	MCF-7	22.5	[77]
			HeLa	21.8	
*Amphipterygium adstringens*	Leaves	Methanol	UACC-62	7.3	[78]
			OVCAR-3	4.4	
			NCI-H460	28	
*Lophophora williamsii*	Cacti	Methanol	C6	1.92	[55]
*Amphipterygium*	Bark	Methanol	HepG2	41.77 ± 6.18	[79]
			Vero	197.98 ± 4.71	
*Cissus incisa*	Leaves	CHCl3/MeOH	HeLa	63 ± 7	[80]
			PC3	43 ± 4	
*Cnidoscolus multilobus* (Pax)	Leaves	Ethanol/water	HeLa	62	[81]
*Capsicum chinense*	Leaves	Methanol	MCF-7	0.38 ± 0.01	[82]
	Stems		MCF-7	2.01 ± 0.33	
	Peduncles		MCF-7	0.46 ± 0.02	
*Semialarium mexicanum (Miers) Mennega*	Root bark	Petroleum ether	MDA-MB-231	55.5	[83]
			MCF10A	66.8	
*Asclepias subulata*	Aerial	Ethanol	HCT-116	0.4	[84]
*Agave lechuguilla* Torr	Leaves	Ethanol	MCF-7	>150	[85]
			HeLa	89	
			Vero	126	
*R. communis*	Aerial	Methanol	Vero	34.8	[86]
		Ether		326.8	
Hippocratea celastroides	Leaves	Ethanol	MCF-7	2.29	[87]
	Stem			2.57	
	Root			2.81	
*Carica papaya* L.	Flower	Ethanol	Vero	62.5	[88]
*Smilax aspera* L	Roots	Acetone	MDA-MB-231	695	[89]
			A549	535	
			OVCAR3	117	
*Argemone mexicana Linn.*	Whole	Methanol	MCF-7	95.50 ± 3.69	[90]
			HeLa	38.01 ± 1.77	
*Phaseolus vulgaris* L.	Bean	Water/methanol	Caco2	81.2	[91]

IC_50,_ inhibitory concentration 50 (concentration needed to inhibit 50% cellular viability); HeLa (cervical carcinoma); L5178Y, lymphoblast cell line; 4T1, stage IV human breast cancer; PC3, stage V prostatic adenocarcinoma; SiHa, carcinoma of the uterus; MCF7, breast cancer; Calu-1, lung epidermoid carcinoma; HepG2, hepatocellular carcinoma; UACC-62, pancreatic cancer metastatic; OVCAR-3, adenocarcinoma of the ovary; NCI-H460, lung carcinoma; C6, the brain of a rat with glioma; Vero, kidney tissue; MDA-MB-231, human breast cancer; MCF10A, fibrocystic breasts; A549, lung cancer; Caco2, colorectal adenocarcinoma.

## 5. Plants in the Treatment of Colorectal Cancer

Colorectal cancer (CRC) is one of the most prevalent and incident cancers worldwide; it is the second most common adult cancer in women, the third most common in men, and the second cause of cancer-related death worldwide. There were over 1.8 million new cases in 2018 in developed countries. In the United States, 148,000 new cases and 53,200 deaths per year are estimated; in Mexico, according to the Mexican Institute of Social Security (IMMS), it is estimated that 15,000 new cases are diagnosed each year. If colorectal cancer is detected in the early stage, it is curable. Therefore, early detection can reduce the mortality rate of colorectal cancer. However, approximately 25% of CRC patients with the advanced-stage disease will develop metastasis at diagnosis [92,93,94]. Different factors are related to cancer incidence, including genetic and environmental factors, and lifestyle factors such as physical inactivity, sedentary behavior, and excessive caloric intake, leading to energy imbalance, progressively leading to obesity. Generally, treating patients with colon cancer includes surgery (segmental colon resection), chemotherapy, and radiotherapy, but some plants are used for the treatment of this disease (Table 2; Figure 2) [95].

*Thalassia testudinum* is a seagrass that forms dense, extensive pastures in the Mexican Caribbean; it presents a dense underground network of roots and rhizomes that serves as a support and storage system for a series of short vertical shoots, each one having two to eight trap-shaped leaf blades, they are the leading food of green turtles. *Chelonia mydas*. *T. testudinum* has been studied for its beneficial properties, such as anti-inflammatory, cytoprotective, antioxidative, and neuroprotective properties related to the high polyphenolic content [96]. Delgado-Roche et al., 2020, studied the activity of the polyphenolic fraction of *T. testudinum* against a colorectal cancer cell line (HCT15) and obtained an IC_50_ value of 22.47 ± 1.30 μg/mL after 48 h of treatment, with ROS overproduction and pro-apoptotic effects [97].

*Annona muricata* is a tropical, fruit-bearing tree of the family *Annonaceae* found in South America; in Mexico, the top producing states are Nayarit, Colima, and Michoacan. *A. muricata*, commonly known as soursop, graviola, or guanabana, has large, glossy, dark green leaves. The aerial parts of guanabana have several applications: fruits have been widely used as a food ingredient, while several preparations, especially decoctions of the bark, fruits, leaves, pericarp, seeds, and roots, have been extensively used in traditional medicine to treat multiple ailments including cancers, diabetes, arthritis, hypertension, snake bite, diarrhea, headache, and malaria [98]. Daddiouaissa et al., 2021, investigated the effect of a pulp extract of guanabana fruit on the metabolomics behavior of colon cancer cells (HT29) using an untargeted GC-TOFMS-based metabolic profiling. Pathway analysis of metabolomic profiles revealed an alteration of many metabolic pathways, including amino acid metabolism, aerobic glycolysis, urea cycle, and ketone body metabolism that contribute to energy metabolism and cancer cell proliferation, with the conclusion that the pulp extract is a promising anticancer agent [99]. Moghadamtousi et al., 2014, investigated the anticancer properties of ethyl acetate extract of *A. muricata* leaves on HT-29 and HCT-116 colon cancer cells. The extract exhibited significant cytotoxic effects on HCT-116 and HT-29 cells with an IC_50_ value of 11.43 ± 1.87 mg/mL and 8.98 ± 1.24 mg/mL against HT-29 and HCT-116 cells, respectively. The mechanistic effect in cancer cells was the excessive accumulation of ROS followed by disruption of MMP, cytochrome c leakage, and activation of the initiator and executioner caspases in both cellular lines [100].

*Opuntia ficus-indica* (L.) Mill. is a dicotyledonous angiosperm tree-like cactus belonging to the family Cactaceae (subfamily Opuntiodeae, Genus Opuntia), also known as prickly pear, and originating from Mexico. However, it can be found in all American hemispheres and grows worldwide, such as in Africa, Australia, and the Mediterranean basin. Several ingredients and bioactive compounds that show pharmacological properties have been identified in the fruit, including carbohydrates; lipids; proteins with their specific amino acids; vitamins; minerals; and phenolic compounds, in particular, phenolic acids (hydroxycinnamic acids and hydroxybenzoic acids); flavonoids; lignins; and stilbenes. This plant is used in traditional medicine to treat diabetes, high cholesterol, obesity, and hangovers. It is also known for its antiviral and anti-inflammatory properties [101,102,103]. Antunes-Ricardo et al., in 2014, evaluated the effect of *O. ficus* alkaline extracts and purified isorhamnetin glycosides (obtained from the extracts) against two different human colon cancer cells (HT-29 and Caco2). The OFI extracts and purified isorhamnetin glycosides were more cytotoxic against HT-29 cells than Caco2 cells. The cytotoxic activity depends on the extraction time and is related to the activation of Caspase 3/7 and apoptosis [104]. In 2021 the authors evaluated the effects of *Opuntia ficus*-indica extract (OFI-E) and its glycoside isorhamnetin-3-O-glucosylrhamnoside (IGR) on an HT29 cell line and the growth of human colorectal adenocarcinoma cells as well as in a xenografted-immunosuppressed mice model. The extract increased apoptosis induction, ROS production, and a G0/G1 cell cycle arrest in HT29 cells. In the mice, the extract and IGR reduced the tumor growth rate through the overexpression of cleaved Caspase-9, Hdac11, and Bai1 proteins and reduced the myeloperoxidase activity and total cholesterol levels [105].

*Punica granatum*, known as cordelina Granada, red pomegranate, or pomegranate, is a small tree from 3 to 6 m high with a semi-woody stem. The leaves are shiny and more prolonged than wide and reddish-green. It has showy red or orange flowers with numerous thread-like stamens. The fruits are globose and red with a kind of crown at one end, with many fleshy, edible seeds that are red. It is characterized by its antioxidant, antihypertensive, and chemopreventive properties in different tissues due to its content of phenolic compounds present in different parts of the plant. For example, the pericarp of the fruit is a rich source of bioactive molecules, such as ellagitannins, polyphenols, flavonoids (luteolin, kaempferol, and quercetin), and anthocyanidins (delphinidin, cyanidin, and pelargonidin); the juice presents a high content of anthocyanins and the leaves and flowers contain tannins and flavones such as apigenin and gallic acid [106,107]. Rodrigues et al., 2020 evaluated the antiproliferative activity of a fast one-step solid–liquid extract of pomegranate grains in HT29 cells; the results showed an IC_50_ value of 318 μg/mL related to the presence of antioxidant compounds such as gallic acid, caffeic acid, catechin, and epicatechin gallate [108].

## 6. Future Perspectives of Medicinal Plant Antioxidants in Cancer Therapy

Cancer is one of the most critical public health problems worldwide due to its high incidence and mortality rate. The principal treatments are chemotherapy and radiotherapy, although both are associated with several side effects because the action to kill cells is not specifically only for cancer cells. Developing an effective treatment with anticancer properties and minor adverse effects is necessary. Currently, it is common for patients to choose a non-traditional anticancer treatment in addition to conventional treatment. Natural compounds with anticancer properties, such as antioxidants, can kill transformed or cancerous cells without being toxic to healthy cells. Antioxidants are bioactive molecules widely distributed in the plant kingdom. They have been used in traditional Mexican medicine since ancient times to treat different sicknesses, including cancer, as observed in the reports cited in this bibliographical review [109,110,111,112]. However, it is essential to ask ourselves what is missing for the plants used in traditional Mexican medicine to be used as a medicine to treat cancer. It is necessary to elucidate the mechanism of action of compounds that eliminate cancer cells, not only in cell lines but also in preclinical and clinical studies in animal models and humans, as well as carrying out toxicological studies of the side effects that may exist after the consumption of these plants. In the same way, it is essential to emphasize that most of the reported studies used concentrated extracts of the plants, which were made with different solvents that may not be allowed for human consumption. In addition, there is the fact that when dealing with living materials such as plants, we may observe a difference in the number of antioxidant compounds, as well as in the cytotoxic activity, depending on the year of harvest, place of planting, and even the climate that has prevailed during its growth. For these reasons, it is of the utmost importance that we carry out studies of the chemical composition of the plants and, if possible, the identification and isolation of the compounds responsible for the activity, which could promote a better understanding of the mechanism of action and the compounds’ potential use as a treatment. Another point to consider is the bioavailability and bioaccessibility studies of the compounds present in plants since, once in the human body, these parameters can decrease, so incorporating these compounds in controlled-release systems would be a great help in allowing their delivery and protecting them from environmental factors. Even with the aforementioned, antioxidant compounds from Mexican medicinal plants represent a viable alternative for developing new treatments against different types of cancer, and it is an area that requires further study.

## 7. Conclusions

This review presents scientific evidence of the antioxidants present in Mexican medicinal plants used for cancer treatment; the information collected suggests that most studies have been carried out using aqueous, ethanolic, and methanolic extracts. They have been tested in cell lines of different types of cancer, and the activity is mainly attributed to the generation of ROS, activation of caspase 3/7, and induction of apoptosis, which is related to the presence of different antioxidant compounds in plants. On the other hand, further efforts are required to clarify and understand the mechanisms of action through which traditional plants and their antioxidant compounds reduce the cellular viability of cancer cells; however, the evidence confirms the role of plants as an excellent source of antioxidant compounds able to fight cancer and other sicknesses and support the use of Mexican medicinal plants.

## Figures and Tables

**Figure 1 pharmaceuticals-16-00482-f001:**
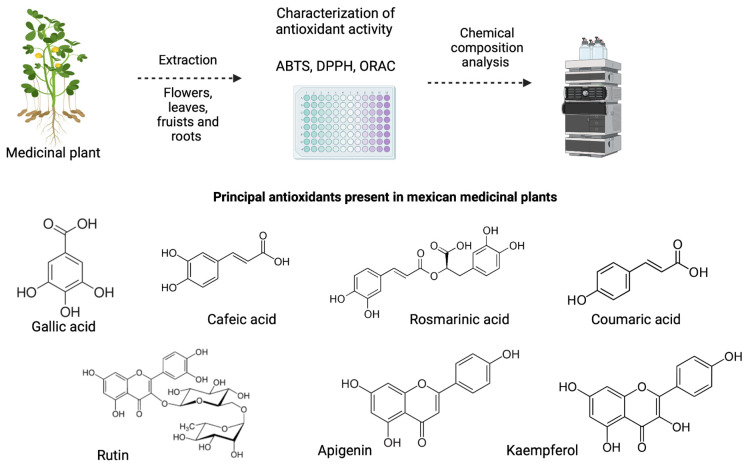
Process of antioxidant extraction and characterization and principal phenolic compound present in Mexican medicinal plants.

**Figure 2 pharmaceuticals-16-00482-f002:**
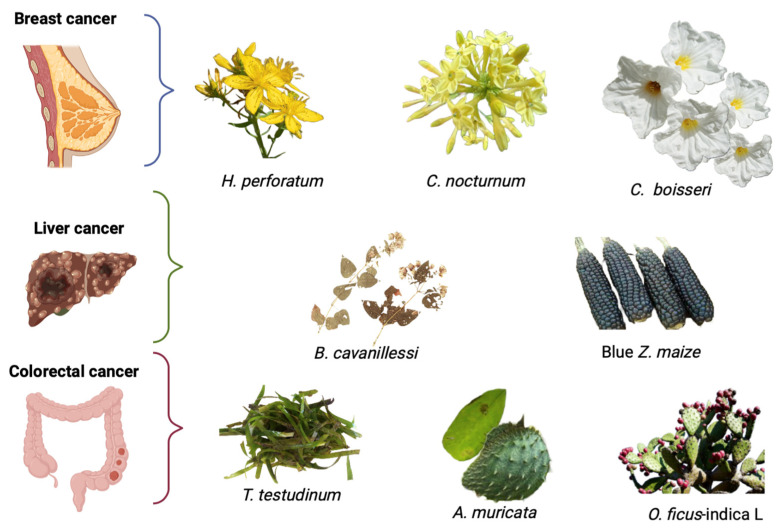
Mexican medicinal plants used for the treatment of different types of cancer. The images were taken from the World Checklist of Selected Plant Families, WCSP record.

## Data Availability

Data sharing not applicable.

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
