# Peer review of "Antioxidants in Traditional Mexican Medicine and Their Applications as Antitumor Treatments"

_pharmaceuticals, 2023, doi:10.3390/ph16040482_

Round 1

Reviewer 1 Report

This review was written quite informative, but I think the section on antitumor treatments and applications should be expanded. Other issues that need to be corrected in the draft of the article are shown in the pdf file. There are typos and repetitions in the bibliography section, as well as incorrect reference numbers in the main text. These need to be carefully rechecked.

Author Response

We greatly appreciate Reviewers #1 for their valuable comments and suggestions. 

Reviewer 2 Report

I believe that the manuscript in its present form is suitable for publication in this journal.
Regards

Author Response

We greatly appreciate Reviewers #2 for their valuable comments and suggestions. 

Reviewer 3 Report

Dear Sir/Madam
Thank you so much for your effort.
Antioxidants in traditional Mexican medicine and their applications as antitumor treatments was studied
The article is well organized and provides significant and up to date studies regarding the use of antioxidants from multiple Mexican plants in the fight against cancer. Few minor corrections are needed.

Line 42: Modify to 'More than 19.3 million new cancer cases were diagnosed and reported recently, leading to approximately 10 million deaths in 2020'

Line 53: Modify to: 'the first causes functional deficiencies'

Line 57: 'these side effects reduce the quality life of the patients

Lines 138-140: Here it should be mentioned that this paragraph refers to Mexican plants, as in general some plants are toxic or not widely available

Line 208: 'BC is diagnosed in the early stages when the tumor can be retracted'

Line 229: The abbreviation of MFLCN (methanolic fraction obtained from ethyl acetate extract of leaf of Cestrum nocturnum) according to Kumar et al. (2022) needs to be mentioned

Line 294: 'Generally is commercially found in herbal stores and is used as a treatment for ulcers'

Author Response

We greatly appreciate Reviewers #3 for their valuable comments and suggestions. 
